# An Optimization Method for PCB Surface Defect Detection Model Based on Measurement of Defect Characteristics and Backbone Network Feature Information

**DOI:** 10.3390/s24227373

**Published:** 2024-11-19

**Authors:** Huixiang Liu, Xin Zhao, Qiong Liu, Wenbai Chen

**Affiliations:** 1School of Automation, Beijing Information Science and Technology University, Beijing 100192, China; liuhx@bistu.edu.cn (H.L.); 2022020422@bistu.edu.cn (X.Z.); liuqionglq@126.com (Q.L.); 2Jiangxi Research Institute of Beihang University, Nanchang 330096, China

**Keywords:** defect detection, PCB, explainability, YOLOv8, target characteristics

## Abstract

Printed Circuit Boards (PCBs) are essential components in electronic devices, making defect detection crucial. PCB surface defects are diverse, complex, low in feature resolution, and often resemble the background, leading to detection challenges. This paper proposes the YOLOv8_DSM algorithm for PCB surface defect detection, optimized based on the three major characteristics of defect targets and feature map visualization. First, to address the complexity and variety of defect shapes, we introduce CSPLayer_2DCNv3, which incorporates deformable convolution into the backbone network. This enhances adaptive defect feature extraction, effectively capturing diverse defect characteristics. Second, to handle low feature resolution and background resemblance, we design a Shallow-layer Low-semantic Feature Fusion Module (SLFFM). By visualizing the last four downsampling convolution layers of the YOLOv8 backbone, we incorporate feature information from the second downsampling layer into SLFFM. We apply feature map separation-based SPDConv for downsampling, providing PAN-FPN with rich, fine-grained shallow-layer features. Additionally, SLFFM employs the bi-level routing attention (BRA) mechanism as a feature aggregation module, mitigating defect-background similarity issues. Lastly, MPDIoU is used as the bounding box loss regression function, improving training efficiency by enhancing convergence speed and accuracy. Experimental results show that YOLOv8_DSM achieves a mAP (0.5:0.9) of 63.4%, representing a 5.14% improvement over the original model. The model’s Frames Per Second (FPS) reaches 144.6. To meet practical engineering requirements, the designed PCB defect detection model is deployed in a PCB quality inspection system on a PC platform.

## 1. Introduction

The application of computer vision in the field of measurement has become increasingly widespread, with approximately 25% of research focusing on vision-based fault and defect diagnosis, detection, and prediction [1]. Among these, small object detection and model lightweighting remain primary challenges in defect diagnosis [2,3]. This issue is especially prominent in printed circuit board (PCB) inspections, where PCBs are widely used in industries such as healthcare, defense, wearable devices, and smartphones (as shown in Figure 1). As automation technology advances, the application of PCBs continues to expand, driving market growth. In 2020, the global PCB market was valued at USD 70.92 billion, and it is projected to reach USD 86.17 billion by 2026, with an annual growth rate of 3.3%. Since PCB quality directly affects device performance, accurately detecting tiny defects is critical. However, traditional detection methods often struggle with identifying such small and diverse defects [4], highlighting the need for high-precision detection.

Traditional PCB surface defect detection methods, such as manual inspection, electrical testing, and infrared detection, are often costly and inefficient, rendering them unsuitable for the industrial scale required for PCB defect detection [5]. To reduce dependency on manual inspection, Automated Optical Inspection (AOI) technology has gained prominence. AOI combines machine vision with high-resolution optical sensors and image-processing algorithms to identify defects on PCB surfaces. Image-processing algorithms in AOI are typically divided into two categories: traditional object detection algorithms that combine image-processing with machine learning, and end-to-end object detection algorithms based on deep learning.

The first category, traditional object detection algorithms, commonly employs a sliding window framework. This approach commences with feature point matching using operators. Subsequently, machine learning algorithms, such as the Adaboost algorithm and Support Vector Machine, serve as classifiers to accomplish the object detection task. For instance, Moganti et al. [6] were among the first to apply machine learning algorithms to PCB defect detection. Ke et al. introduced the PCA-SIFT [7] algorithm, incorporating Principal Component Analysis (PCA) to enhance matching efficiency. Moreover, Bay et al. developed the SURF [8] algorithm, which constructs a Hessian matrix for key point localization, significantly reducing computation and increasing detection speed. In addition, Rublee et al. introduced the ORB [9] algorithm, leveraging Fast [10] for key point detection and Brief [11] for computing feature descriptors, achieving detection speeds surpassing SIFT and SURF. Kaur et al. [12] utilized image differencing operations for defect identification on bare PCBs, effectively detecting bare board defects. Furthermore, Baygin et al. [13] proposed a non-contact defect detection algorithm, extracting features from the test image and comparing it to a reference image to detect PCB through hole defects. In summary, the first class of object detection algorithms, relying on handcrafted features and template matching, is both time-consuming and costly.

Currently, deep learning-based end-to-end object detection algorithms are mainly categorized into two-stage and one-stage detection algorithms. Two-stage algorithms involve initial region proposal generation followed by object detection within these proposed regions. Notable two-stage algorithms include R-CNN [14], Faster R-CNN [15], and SPP-Net [16]. In contrast, one-stage algorithms perform object detection in a single step by using the entire input image, typically employing a single CNN for both region proposal and object detection. Prominent one-stage detection algorithms include the You Only Look Once (YOLO) series [17], Single Shot-multiBox Detector (SSD) [18], CenterNet [19], and DETR [20].

Ding et al. [21] developed a Tiny Defect Detection Network (TDD-Net) technique specifically for tiny PCB defects. This network combines the backbone network of the Faster R-CNN algorithm with a multi-scale pyramid, also known as Feature Pyramid Networks (FPN), facilitating information exchange across different feature layers and thereby improving the accuracy of PCB defect detection. Moreover, Li et al. [22] conducted hybrid training of the Faster R-CNN and YOLOv2 algorithms, integrating the detection results from both models. The proposed technique achieved a precision rate of 96.73% in real-world tests. Fung et al. [23] proposed two new PCB defect detection techniques, namely Selective Feature Attention (SF Attention) and Pixel Shuffle Pyramid (PSPyramid), for identifying and integrating important features. Experiments showed that their approach reduced the AP50 error by 16% compared to existing methods on the TDD dataset, significantly improving defect detection performance for PCB quality assurance. Zeng et al. [24] proposed a novel enhanced multi-scale feature fusion method (ABFPN), which utilizes dilated convolution operators with different dilation rates to fully leverage contextual information. The algorithm [21,22,23,24] achieved significant improvements in accuracy. However, the two-stage framework led to a complex model and large parameter count, which is not ideal for real-time detection in PCB production environments.

Tang et al. [25] improved the YOLOv5 model and proposed the PCB-YOLO technique. This network utilizes the K-means++ algorithm to redefine anchor box sizes and includes a specialized layer for small object detection, significantly enhancing the algorithm’s capability to detect small-sized targets. Additionally, the integration of a joint attention mechanism and Swin Transformer modules in the backbone network reinforces the capture of channel and spatial information. Consequently, experimental results showed an average precision (mAP) of 95.97% and an FPS of 92.5. Yu et al. [26] proposed an efficient Scale-aware Network (ES-Net), which improves overall defect detection by addressing issues such as information loss in detecting small objects and mismatch between detection head receptive fields and target sizes. Yang et al. [27] introduced a novel detection network for PCB defect detection based on Coordinate Feature Refinement (CFR). The CFR structure aims to suppress conflicting information from different levels to highlight PCB defect features. Du et al. [28] proposed an improved YOLOv5s network, called YOLO-MBBi, for detecting surface defects on PCBs. This network employs the Mobile Inverted Residual Bottleneck Blocks (MBConv), CBAM attention, BiFPN, and depthwise convolutions to replace layers in the YOLOv5s network. Therefore, they applied the SIoU loss function instead of CIoU during training. The experimental results showed that YOLO-MBBi achieved a mAP (0.5) of 95.3% and an FPS of 48.9. Yuan et al. [29] proposed a novel YOLOHorNet-MCBAM-CARAFE (YOLO-HMC) network based on an improved YOLOv5 framework, which can accurately and efficiently identify small-sized PCB defects with fewer model parameters, while algorithms [25,26,27,28] showed improvements in the mAP (0.5) precision evaluation metric, there is still room for improvement in terms of model size, computational complexity, and inference speed. Moreover, these related works have not explored enhancements in the more comprehensive precision evaluation metric, mAP (0.5:0.9). The algorithm [29] achieved significant improvements in model size, computational complexity, and inference speed, contributing to the enhancement of the comprehensive precision evaluation metric, mAP (0.5:0.9). However, there remains room for improvement in these aspects.

To address the challenges in PCB defect detection, this paper enhances YOLOv8 by incorporating defect-specific target features and feature map visualization, resulting in the proposed YOLOv8_DSM—a model optimized for detecting small PCB defects. The algorithm introduces deformable convolution technology in the form of CSPLayer_2DCNv3, which enhances the backbone network’s adaptive feature extraction capability to accommodate diverse defect patterns. Additionally, based on backbone feature map visualization, the SLFFM is designed, integrating SPDConv downsampling with a BRA dynamic sparse attention mechanism to effectively extract detailed features and enhance long-range modeling capabilities. Finally, the Minimum Point Distance-based IoU (MPDIoU) loss is used to improve training efficiency and accuracy during bounding box regression. Experimental results show that YOLOv8_DSM increases mAP (0.5:0.9) to 63.4%, with an FPS of 144.6. The main contributions of this paper are as follows:To address the complexity and diversity of PCB surface defects, deformable convolution is introduced to design the CSPLayer_2DCNv3 convolution module. This design reduces parameter and memory complexity while enhancing the model’s adaptive feature extraction ability, effectively extracting diverse defect features. The combination of different DCNv3 layers provides rich feature diversity.To tackle the small area ratio and the similarity of PCB surface defects to the background, SLFFM is designed based on backbone feature map visualization. It incorporates separation downsampling convolution (SPDConv) and the BRA mechanism to retain fine-grained shallow features, effectively filter out noisy information, reduce computational costs, and improve the model’s long-range modeling capabilities.To enhance bounding box regression accuracy, the Minimum Point Distance-based IoU (MPDIoU) loss is adopted as the regression loss function, resulting in improved convergence speed and regression accuracy.A user-friendly PCB quality inspection system was designed based on the PCB surface defect detection algorithm proposed in this paper.

## 2. Methods

### 2.1. Selection of the Baseline Model

We collected performance data from four YOLO algorithms (YOLOv5n, YOLOv6n, YOLOv7-tiny, and YOLOv8n) on the COCO2017 object detection dataset, sourced from official open-source GitHub projects. These models were evaluated at an input size of 640 × 640 for their parameter count, computational demand, and multi-scale accuracy metric mAP (0.5:0.95), detailed in Table 1. Results show that YOLOv5n, despite having the lowest parameters (1.9 M) and FLOPs (4.2 G), scored only 28.0 in mAP, indicating weaker performance. In contrast, YOLOv6n, with increased parameters (4.7 M) and FLOPs (11.4 G), achieved the highest mAP of 37.5 among the models. YOLOv7-tiny had parameters of 6.2 M and FLOPs of 13.1 G, with an mAP of 36.8, slightly lower than YOLOv6n. Notably, YOLOv8n managed a balance between computational resource consumption and detection accuracy with parameters of 3.2 M and FLOPs of 8.7 G, achieving an mAP of 37.3, close to the highest. Overall, YOLOv8n offers an ideal balance between performance and efficiency with near-optimal mAP scores at lower computational costs, making it the best choice.

### 2.2. YOLOv8 Detection Algorithm

YOLOv8 is a single-stage object detection algorithm proposed by Ultralytics in 2023, built upon YOLOv5. It merges the most advanced State-Of-The-Art (SOTA) technologies with new functionalities and improvements [30]. YOLOv8 has a wide range of applications in the field of object detection [31,32]. This algorithm comes in five variants: YOLOv8n, s, m, l, and x. The complexity and computational demand of each model increase with its network depth and width, allowing users to select a network structure based on their specific application needs. More specifically, YOLOv8n is optimized for NVIDIA Jetson Nano devices, offering relatively fast detection speeds and high accuracy while being applied on embedded, low-cost, and low-power devices [33]. The architecture of YOLOv8n, proposed in Figure 2, primarily consists of three parts: Backbone, Neck, and Detect. These components are discussed next.

#### 2.2.1. Backbone

YOLOv8’s backbone is a modified CSPDarknet53, down-sampling input features five times into scales P0-P4, shown in Figure 2a. It replaces Cross-Stage Partial (CSP) modules with CSPLayer_2Conv modules (as shown in Figure 2d), having two bottleneck block types. When ’add’ is true, bottlenecks add residual shortcuts like resnet blocks; otherwise, they stack two convolutions. The CSPLayer_2Conv module merges Efficient Layer Aggregation Network (ELAN) [34] and CSPNet benefits, using gradient bifurcation connections for better information flow and a lightweight structure. The CBS module performs convolution, batch normalization, and SiLU activation (as shown in Figure 2e). Lastly, the Spatial Pyramid Pooling Fast (SPPF) module pools feature maps into fixed-size mappings for adaptive size output (as shown in Figure 2c).

#### 2.2.2. Neck

YOLOv8 has implemented a PAN-FPN structure in its neck, as illustrated in Figure 2b. The initial step involves the multilevel feature fusion through FPN, combining deep and shallow features. This ensures the model retains robust semantic information while preserving more image details, effectively addressing issues related to scale variation. However, this process may lead to some loss of target location information. To mitigate this, PAN-FPN introduces another bottom-up feature fusion method, PAN, enhancing the learning of positional information.

#### 2.2.3. Detect

In prior YOLO series models, predictions for classification and bounding box regression on the input feature maps occurred simultaneously, sharing parameters from the preceding layer. However, YOLOv8 adopts a different approach by separating these predictions. This decoupling of prediction heads has been shown to accelerate model convergence and improve detection accuracy.

### 2.3. The Proposed Method

YOLOv8n still has significant room for improvement in accurately detecting PCB defects with complex shapes and low-resolution features that blend into similar backgrounds. Therefore, leveraging specific defect characteristics and backbone network feature maps, we enhanced YOLOv8 to create YOLOv8_DSM, tailored for precisely detecting small-target PCB defects. As shown in Figure 3, we adapted the CSPLayer_2Conv by integrating deformable convolutions to form CSPLayer_2DCNv3, which accommodates the diversity of PCB defects. Additionally, we introduced the SLFFM module to better preserve detail during feature extraction. Moreover, we employed the MPDIoU function to refine bounding box regression training, significantly enhancing detection accuracy and speed.

#### 2.3.1. CSPLayer_2DCNv3 Module

CNNs struggle with capturing long-distance dependencies and learning from large-scale, complex data when compared to models based on Multi-Head Self-Attention (MHSA). To address the gap between convolution and MHSA, a direct method involves introducing long-range dependencies and adaptive spatial aggregation into regular convolution. Consequently, in the literature [35], the authors re-engineered DCNv2 to create a new convolution variant, known as Deformable Convolution DCNv3, whose structure is illustrated in Figure 3g.

DCNv3 expands upon DCNv2 by incorporating shared weights among convolutional neurons, introducing a multi-group mechanism, and employing the softmax normalization function based on sample points. Based on the modifications from DCNv2, given an input $x\subseteq R^{C\times H\times W}$ and a current pixel $p_{0}$, DCNv3 is calculated as shown in Equation (Equation 1).
(1)$$y\left(p_{0}\right)=\sum_{g=1}^{G}\sum_{k=1}^{K}w_{g}m_{gk}x_{g}\left(p_{0}+p_{k}+\Delta p_{gk}\right)$$
where *G* represents the total number of aggregation groups. For the $g^{th}$ group, $w_{g}\in R^{C\times C^{\prime }}$ denotes the group’s location-irrelevant projection weights, and $C^{\prime }=C/G$ represents the group dimension. In addition, $m_{gk}\in R$ signifies the modulation scalar for the $K^{th}$ sampling point in the $g^{th}$ group, normalized using the softmax function along the *K* dimension. As for $x_{g}\in R^{C^{\prime }\times H\times W}$, it denotes the sliced input feature map. Finally, $\Delta p_{gk}$ refers to the offset corresponding to the grid sampling positions $p_{k}$ of the $g^{th}$ group.

DCNv3 overcomes the limitations of traditional convolution and attention methods by efficiently handling long-distance dependencies and adaptive spatial aggregation, improving performance in complex visual tasks. It retains the inductive convolution bias, allowing efficient learning with limited data and training time. With sparse sampling, DCNv3 is computationally and memory efficient compared to previous methods, using 3 × 3 kernels to learn long-range dependencies. This simplifies optimization and avoids complexities of large kernel technologies, like reparameterization. Consequently, DCNv3 significantly enhances efficiency, reduces resource use, and boosts model performance.

The complexity and diversity of PCB surface defects, particularly rat bite defects with their varied types and shapes, necessitated a reconstruction of the CSPLayer_2Conv module through the integration of DCNv3. This updated module, named CSPLayer_2DCNv3, enhances the original Bottleneck structure in CSPLayer_2Conv by incorporating DCNv3 convolution, which allows for deformable convolution of the input feature map via a flexible sampling grid. Additionally, offsets derived from an extra convolution layer assist in accurately learning the scale, pose, viewpoint, and deformation of the target object. This modification significantly improves the effectiveness of feature information processing during gradient diversion. The architecture of CSPLayer_2DCNv3 is elaborated in Figure 3e,f.

As a result, CSPLayer_2DCNv3 not only reduces model parameters, conserving computational resources, but also possesses the capability to adaptively extract features of PCB defects and engage in long-range modeling. The addition of the deformable convolution module enhances the model’s flexibly in handling issues related to insufficient receptive fields for detection points corresponding to small targets. This adaptation improves focus on detecting small objects and effectively reducing false negatives and false positives, thereby improving detection accuracy.

#### 2.3.2. Shallow Low Semantic Multi-Scale Fusion Module

In PCB surface defect detection, defects are typically small and sparse, resulting in low-resolution, low-level feature information. This is compounded by the visual similarity between defects and the background, particularly in multi-copper and burr defects. To enhance detection of such small-scale defects, the SLFFM module was developed, with its detailed structure shown in Figure 3b. To ensure effective integration of detailed feature information, heat maps of feature maps from all CSPLayer-2Conv in the backbone were analyzed, as shown in Figure 4. This visualization helps in understanding how the SLFFM module can improve the detection of nuanced PCB defects. The heat map analysis of the backbone’s second and fourth layers revealed substantial defect feature information. However, the second layer’s features were not initially integrated into the PAN-FPN. To address this issue, the SLFFM module incorporates the feature map from the second layer of the backbone, which contains a substantial amount of fine-grained feature information. SPDConv [36] is used for down-sampling, preserving shallow, detailed features and enriching the P2 feature map in YOLOv8’s small object detection head with essential defect-related shallow features. This enhances the detection of small-scale PCB defects. During the module’s multi-scale feature integration, the dynamic sparse attention mechanism of BRA [37] filters out irrelevant features at a coarse level, thereby emphasizing shallow semantic information of defect features, such as defect shapes. This approach not only preserves critical detailed features but also strengthens the model’s long-range capture capabilities through the global attention mechanism.

SPDConv consists of a Space-to-Depth (SPD) layer and a non-stride convolution layer. SPDConvemploys an (original) image transformation technique to perform down-sampling on the feature maps within and across the entire CNN, as illustrated in Figure 5.

The SLFFM method enhances PAN-FPN input by adding richer, learnable details from the fourth layer of the backbone, unlike the original design. It necessitates down-sampling of larger scale features for effective fusion across different levels. To preserve detailed information in the features from the fourth backbone layer, SLFFM uses SPDConv as its down-sampling method.

The complexity of the PCB surface defect detection environment, coupled with the similarity of defect features to the background in areas like multi-copper and burr defects, necessitates a detection model that focuses on key information in input features, acquires global spatial information from the feature maps, and enriches contextual semantic information. Therefore, we introduced the BRA, in the SLFFM. After feature aggregation, BRA is deployed for feature learning and the extraction of the integrated features. BRA applies query adaptability to first filter out the least relevant key-value pairs in the coarse grained regions of the input feature map, effectively finding highly relevant key-value pairs, and performing attention computation. This significantly reduces the computational cost as well as the storage consumption and enhances the model’s perception of the input content, as illustrated in Figure 6.

Referring to Figure 6a, the input feature map $X\in R^{H\times W\times C}.$ is first divided into S×S sub-regions, each containing $\frac{HW}{S^{2}}$ feature vectors. This step reshapes the original feature map *X* into $X^{r}\in R^{S^{2}\times \frac{HW}{S^{2}}\times C}$. Consequently, the feature vectors are transformed linearly to obtain the query matrix *Q*, key matrix *K*, and value matrix *V*. The calculation process is as per Equations (Equation 2)–(Equation 4):(2)$$Q=X^{r}W^{Q}$$
(3)$$K=X^{r}W^{K}$$
(4)$$V=X^{r}W^{V}$$
where $W^{\varrho },W^{K},W^{V}$ represent the projection weights for the query, key, and value, respectively.

Consequently, an attention relationship between regions is established by constructing a directed graph to locate the relevant regions for a given region. Perform regional averaging on *Q* and *K* of each region to obtain the regional level $Q_{r}$ and $K^{r}\in R^{S^{2}\times C}$. Then, compute the dot product of $Q_{r}$ and $K_{r}$ to obtain the adjacency matrix $A^{r}\in R^{S^{2}\times S^{2}}$ to measure the degree of association between regions. Therefore, the calculation process for $A^{r}$ is represented in Equation (Equation 5):(5)$$A^{r}=Q^{r}{\left(K^{r}\right)}^{T}$$

Subsequently, $A^{r}$ is pruned to filter out the least relevant tokens in $A^{r}$ and retain the top *k* most relevant regions in $A^{r}$, resulting in the routing index matrix $I^{r}\in N^{S^{2}\times k}$. The calculation process for $I^{r}$ is represented in Equation (Equation 6):(6)$$I^{r}=topkIndex\left(A^{r}\right)$$

Therefore, the $i^{th}$ row of matrix $I^{r}$ contains the *k* indices of the most relevant regions for the $i^{th}$ region. Subsequently, at the fine-grained level, token-to-token attention computation is performed for the queries in region *i* but it is just applied towards the *k* routed regions. The indices of these *k* regions are denoted as $I_{\left(i,1\right)}^{r},I_{\left(i,2\right)}^{r},\ldots ,I_{\left(i,k\right)}^{r}$. All *K* and *V* tensors from these *k* regions are collected to obtain $K^{g}$ and $V^{g}\in R^{S^{2}\times \frac{kHW}{S^{2}}\times C}$. The computation is displayed in Equations (Equation 7) and (Equation 8):(7)$$K^{g}=gather\left(K,I^{r}\right)$$
(8)$$V^{g}=gather\left(V,I^{r}\right)$$

Finally, the collected $K^{g}$ and $V^{g}$ undergo attention processing, and a Local Context Enhancement LCE(V) term [30] is added to obtain the output tensor *O*. The calculation of this parameter are displayed in Equation (Equation 9):(9)$$O=Attention\left(Q,K^{g},V^{g}\right)+LCE\left(V\right)$$

The BRA is based on a dual-layer routing attention design, as shown in Figure 6b. In this module, DWConv represents the depthwise separable convolution, which can reduce the number of parameters and computational load of the model. Moreover, LN stands for Layer Normalization, accelerating the training and enhancing the model’s generalization ability. In addition, MLP represents a Multi-Layer Perceptron that further processes and adjusts attention weights, enhancing the model’s focus on different features. The "+" sign, denoted in Figure 6b, indicates the concatenation of two feature vectors.

This paper integrates the BRA into the SLFFM module. It applies BRA’s dynamic sparse attention mechanism for feature learning and extraction after stacking features from different layers. BRA not only accounts for the real-time aspect of the detection model but also enhances the model’s focus on critical target information, optimizing its detection performance.

#### 2.3.3. Improved Loss Function

The loss function of YOLOv8 is divided into the binary cross-entropy loss for classification and the bounding box regression loss function $L_{\mathrm{CIo}U}$ [38]. Both elements are added together with the different weights to calculate the total loss. The CIoU loss defines the aspect ratio as a relative value rather than an absolute one. Considering the advantages included in $L_{GIoU}$, $L_{DIoU}$, $L_{\mathrm{CIo}U}$, and $L_{EIoU}$, Siliang et al. [39] proposes a new loss function $L_{MPDIOU}$ for bounding box regression, which is more efficient and accurate. Therefore, $L_{\mathrm{CIo}U}$ in YOLOv8 is replaced by a better-performing $L_{MPDIoU}$.

During the training phase, the loss function is minimized to make the values of the predicted bounding box $B_{prd}={\left[x^{prd},y^{prd},w^{prd},h^{prd}\right]}^{T}$ closely linked to the values of the true bounding box $B_{gt}={\left[x^{gt},y^{gt},w^{gt},h^{gt}\right]}^{T}$.
(10)$$L=\min_{\Theta }\sum_{B_{gt}\in B_{gt}}L\left(B_{gt},B_{prd}|\Theta \right)$$
where $B_{gt}$ represents the set of real annotated bounding boxes, and Θ indicates the parameters of the deep model used for regression. It is important to mention that the typical loss function L adopts the $l_{n}$ norm.

The new IoU-based metric, MPDIoU, aims to minimize the distance between the predicted and ground truth bounding box’s top-left and bottom-right corner points. It simplifies similarity comparisons between bounding boxes and is adaptable for regression, regardless of whether the boxes overlap.

After calculating the coordinates of the upper left and lower right points of the predicted box $B_{prd}=\left[x_{1}^{prd},y_{1}^{prd},x_{2}^{prd},y_{2}^{prd}\right]$ and the real box $B_{gt}=\left[x_{1}^{gt},y_{1}^{gt},x_{2}^{gt},y_{2}^{gt}\right]$, the MPDIoU metric is computed as follows:(11)$$d_{1}^{2}={\left(x_{1}^{gt}-x_{1}^{prd}\right)}^{2}+{\left(y_{1}^{gt}-y_{1}^{prd}\right)}^{2}$$
(12)$$d_{2}^{2}={\left(x_{2}^{gt}-x_{2}^{prd}\right)}^{2}+{\left(y_{2}^{gt}-y_{2}^{prd}\right)}^{2}$$
(13)$$MPDIoU=\frac{A\cap B}{A\cup B}-\frac{d_{1}^{2}}{w^{2}+h^{2}}-\frac{d_{2}^{2}}{w^{2}+h^{2}}$$
where *w* and *h* denote, respectively, the width and height of the input image.

Moreover, the loss function based on MPDIoU is represented in Equation (Equation 14):(14)$$L_{\mathrm{MPDIo}U}=1-\mathrm{MPDIo}U$$

This paper substitutes the bounding box regression loss function with the IoU loss $L_{MPDIoU}$ based on the minimum point distance to address the issues with $L_{CIoU}$ loss, achieving faster convergence speed and more accurate regression results.

## 3. Experiments and Discussion

To evaluate YOLOv8_DSM’s performance in PCB surface defect detection and its enhanced detection module, extensive experiments were conducted. Section 3.1 describes the experimental environment and dataset, while Section 3.2 introduces performance metrics. Section 3.3 analyzes the results in detail. Comparative experiments show YOLOv8_DSM’s superiority over other models, and ablation studies affirm the design’s rationality and effectiveness. These results validate the model’s superiority and provide insights for PCB defect detection research.

### 3.1. Experimental Environment and Data

The experiment utilized an AMD EPYC 9654 96-core processor and two NVIDIA GeForce RTX 4090 graphics cards with 24 GB VRAM each, running on Ubuntu20. The deep learning framework selected for the experiment was Pytorch1.11.0.Training settings included a batch size of 32 and 640 × 640 pixel input images. The optimizer was Stochastic Gradient Descent (SGD), with a learning rate of 0.02, decay momentum of 0.0005, and momentum of 0.937. An early stopping mechanism was used to prevent overfitting and conserve computational resources. Details are provided in Table 2.

The PCB defect dataset was obtained from the PKU-Market-PCB dataset, made publicly available by Peking University’s Intelligent Robot Development Laboratory (available at https://robotics.pkusz.edu.cn/resources/dataset/, accessed on 20 September 2023). It includes 1386 images with an average size of 2777 × 2138 pixels, covering six defect types: missing holes, mouse bites, open circuits, short circuits, burrs, and spurious copper. The limited sample size in the original dataset poses challenges like low detection accuracy, poor robustness, and overfitting during training.

To overcome the shortage of training samples, the original images were augmented, following the method in reference [21]. This increased the dataset to 5334 images through random flipping, rotation, cropping, and cutting, each averaging 600 × 600 pixels. The dataset was split into training, validation, and test sets in a 6:2:2 ratio, providing ample data for better evaluation of the model’s generalization ability. A detailed statistical analysis of the training dataset was performed for effective PCB surface defect detection. The analysis revealed that the average defect size in the training set is 30 × 30 pixels, accounting for about 0.25% of the total image area, categorizing PCB defects as small object defects. Figure 7 illustrates the distribution of different defect types in the training set.

### 3.2. Evaluation Metrics

To test the detection performance of our proposed improved model, we use, as evaluation metrics, the mAP (0.5:0.95), the number of model parameters, the computational complexity, and the detection speed. mAP (0.5:0.95) denotes the mean average precision when the IoU threshold of the detection model is set from 0.5 to 0.95, using an incremental step of 0.05. When the value of mAP (0.5:0.95) is high, the algorithm’s detection results are very accurate across a range of thresholds, providing broad coverage and adaptability to different scenes and application requirements.

### 3.3. Experimental Results and Analysis

This paper thoroughly evaluates the YOLOv8_DSM algorithm performance in PCB small target defect detection. The evaluation combines comparative experiments and ablation studies. For comparative experiments, classic one-stage object detection algorithms were used as benchmarks. These tests were conducted on the same hardware platform and environment to assess YOLOv8_DSM performance advantages. The ablation studies involved the systematic removal or modification of key modules in the algorithm to assess their impact on overall performance. Both the comparison and ablation experiments were consistent, including the use of the same training set, data augmentation strategies, evaluation metrics, and test set. This approach ensures the comparability and credibility of the experimental results. Through these experiments, the algorithm’s performance can be comprehensively evaluated, establishing its superiority relative to other algorithms. Moreover, the effectiveness of each key module can be verified.

#### 3.3.1. Comparative Experiment

In the comparative experiment, to accurately evaluate the performance of our developed model, we conducted an in-depth comparison of its detection capabilities with multiple advanced models. Yolov5n [40], Yolov6n [41], Yolov7-tiny [42], and Yolov8n [30] are classic and high-performing single-stage algorithms that serve as baseline models in numerous studies, while YOLO-MBBi [25], GCC-YOLO [43], Light-PDD [44], and YOLO-HMC [29] are current mainstream algorithms that demonstrate excellent performance in PCB defect detection tasks. The comparison results are shown in Table 3. Our model is specifically designed for targeted industrial applications, focusing on PCB surface defect detection. We meticulously compared each model’s detection accuracy and real-time performance on a PCB defect dataset. Through the experimental data, we aim to provide a comprehensive and rigorous performance comparison, not only showcasing the superior performance of our model but also highlighting its potential value in actual industrial applications.

According to Table 3, our model significantly outperforms other models in the mAP (0.5:0.95) performance metric, achieving the highest detection accuracy in the table at 63.4%. In terms of the number of parameters, our model has 3.8M, which is comparable to YOLOv8n, yet it exhibits a notable improvement in performance. Compared to models with a larger number of parameters, such as YOLO-MBBi and GCC-YOLO, our model not only has fewer parameters but also substantially higher detection accuracy. For instance, YOLO-MBBi and GCC-YOLO have parameters of 8.8M and 8.2M, respectively, but their mAPs are only 52.5% and 48.5%. This demonstrates that our model maintains a lower parameter count while enhancing detection efficiency and accuracy through optimized algorithms, making it particularly suitable for high-precision object detection in resource-constrained environments.

#### 3.3.2. Ablation Experiment

The CSPLayer_2DCNv3 module is primarily designed to effectively address the complexity and diversity of PCB surface defects, especially the various types and shapes of rat bite defects. To explore the impact of varying the number and placement of CSPLayer_2DCNv3 modules replacing the CSPLayer_2Conv in the backbone network on the feature extraction capability of the model’s backbone, we established three different configurations. The baseline, referred to as Backbone, maintains the four layers of CSPLayer_2Conv unchanged. Scheme A replaces all four layers of CSPLayer_2Conv in the Backbone with CSPLayer_2DCNv3. Scheme B replaces the top three layers of CSPLayer_2Conv in the Backbone with CSPLayer_2DCNv3. Scheme C replaces the bottom three layers of CSPLayer_2Conv in the Backbone with CSPLayer_2DCNv3.

According to Table 4, a detailed analysis of the performance across different backbone structures revealed that Scheme C achieved the highest overall accuracy. In contrast, Schemes A and B exhibited a notable decline in accuracy compared to the original backbone. Specifically, Scheme C showed only a slight drop in average accuracy for spur, likely due to a slight reduction in the feature extraction ability for fine details resulting from lower parameter count and computation. However, Scheme C achieved accuracy improvements of 3.23%, 3.43%, and 1.01% for holes, rat bites, and burrs, respectively. Additionally, Scheme C saw an overall increase in average accuracy across all categories. With parameter count reduced to 2.8M and FLOPs lowered to 7.7 G, Scheme C demonstrated reduced model complexity relative to the original backbone. Overall, Scheme C displayed superior accuracy, underscoring its effectiveness in identifying a range of defect types.

We visualized the fourth layer feature maps of Scheme C and the original Backbone using heat maps to demonstrate their capability in extracting features for six different defect types, as shown in Figure 8. The visualization reveals that Scheme C exhibits enhanced feature extraction abilities across all defect categories compared to the original Backbone. To sum up, Scheme C not only reduces model complexity but also enhances feature extraction capabilities, making it particularly suitable for PCB surface defect detection in resource-constrained environments.

The main focus of the SLFFM module is the application of SPDConv for down-sampling and the adoption of the BRA for extracting aggregated feature information. To investigate the impact of SPDConv and BRA on the performance of the shallow low-semantic multi-scale fusion module, an ablation experiment is conducted. The original configuration of the SLFFM module is applied as the baseline. By replacing SPDConv with traditional convolution for down-sampling, the performance of SPDConv in the shallow low-semantic multi-scale fusion module was evaluated. Additionally, the impact of not utilizing the BRA sparse attention mechanism for processing the aggregated features is examined to understand its impact on the module’s performance.

In the ablation experiments of the SLFFM, we compared three different configurations. As shown in Table 5, excluding SPDConv resulted in a 0.2% decrease in accuracy, confirming the critical role of SPDConv downsampling in enhancing performance. When BRA was omitted, the overall average accuracy dropped to 61.8%, a reduction of 1.61%. The absence of the BRA sparse attention mechanism led to performance declines across all categories, with the most significant decrease observed in short circuits. Figure 7 illustrates that short circuit defects have the smallest area among all defect categories, highlighting the substantial impact of the BRA sparse attention mechanism on the model’s ability to detect small targets. Incorporating both SPDConv and BRA enabled the model to achieve high accuracy across all categories, with an average accuracy of 62.8%. This indicates that the combination of SPDConv and the BRA sparse attention mechanism is highly effective in shallow low-semantic multi-scale fusion, collaboratively enhancing the overall performance of the model. The primary contributions of the SLFFM are as follows:Multi-layer Feature Fusion: By integrating the feature maps from the third layer output of the Backbone network with the 80 × 80 feature map of P2, the model’s ability to capture detailed information of small target defects is significantly enhanced, as shown in detail in Figure 4.Effective Retention of Detail Information: Utilizes SPDConv convolution as a down-sampling method combined with a dynamic sparse attention mechanism for sparse sampling. This effectively holds learnable fine-grained shallow features, aiding in the prominent expression of detailed features and improving the model’s accuracy in recognizing minute defects.Enhanced Long-Range Modeling Capability: Enhances the model’s long-range modeling capability. It helps the model to capture global spatial information of the feature map, elevating the contextual semantic information on the feature map. This contributes to identifying defect features from similar background features.

Next, we will explore in detail the impact of various combinations of the DCNv3, SLFFM, and MPDIoU modules on model performance by examining them. The comparison benchmark consists of the YOLOv8 model, providing a performance baseline. We introduce the DCNv3 module, the SLFFM module, and the MPDIoU module each one alone. The addition of each module aims to assess its contribution to the overall model performance, particularly in terms of accuracy and efficiency. By comparing the performance of different configurations, this ablation study identifies the role of each individual module and their combinations in improving the overall functionality of the YOLOv8_DSM model.

Referring to Table 6, the performance of the YOLOv8 model slightly improved to 60.7% after incorporating the DCNv3 module, while the number of parameters and FLOPs decreased to 2.8 M and 7.6 G, respectively. This reveals that DCNv3 can slightly enhance performance while improving efficiency. When the SLFFM module is introduced independently, there is a significant performance improvement, reaching 62.8%; however, the number of parameters also increased to 4 M as well as the FLOPs who reached 11.5 G. This demonstrates that SLFFM significantly enhances performance while increasing the model complexity. Combining the DCNv3 and SLFFM modules further improves performance, reaching an accuracy of 63.1%, with a reduction in the number of parameters and FLOPs to 3.8 M and 11.1 G, respectively. This combination features a synergistic effect in enhancing performance and controlling complexity. Finally, after the combination of the three modules, the mAP (0.5:0.95) reaches its highest value of 63.4%, while keeping the same number of parameters and FLOPs. This indicates that the MPDIoU module further enhances performance without increasing computational burden. To sum up, these experimental results clearly demonstrate that a balanced combination of modules can effectively yield a balance between enhancing model performance and maintaining computational efficiency.

## 4. Defect Detection Model Deployment and Quality Inspection System

We deployed the PCB defect detection model designed in this paper on a PC and developed an intuitive, user-friendly quality inspection system. The system enhances the user experience and work efficiency through features like visual display of detection results and database management.

The quality inspection interface is designed with multiple functional areas to meet the user’s detection needs. First, the input mode selection area is located at the top of the interface, where users can choose between photo, video, and camera inputs. The system processes defect information based on different sources accordingly. The single sample defect detection information display area is located in the center of the interface, showing the detected defect information, including defect type, location, and description, to help users quickly understand the detection results and take appropriate actions. Meanwhile, the detection statistics display area is at the core of the interface, providing statistical information on the detection process, such as the status of the detected boards and detection time. is area allows users to monitor the detection process and ensure accuracy. The system interface is shown in Figure 9.

## 5. Conclusions and Prospects

This paper presents improvements to the one-stage object detection model YOLOv8, based on three key features of defect targets and feature map visualization, resulting in an efficient PCB surface defect detection algorithm, YOLOv8_DSM. The proposed algorithm significantly enhances both accuracy and real-time performance, particularly in detecting defects with complex shapes and small sizes. Firstly, integrating deformable convolution, the reconstructed CSPLayer_2DCNv3 module effectively addresses the diverse features of PCB surface defects, enhancing the model’s adaptive feature extraction capability while maintaining efficiency and reducing computational and storage requirements. Secondly, the introduction of the SLFFM module, combined with the dynamic sparse attention mechanism (BRA), allows the model to better filter noise and retain key features, significantly improving the detection accuracy for small-area defects with detailed similarities. Finally, the use of MPDIoU—a loss function based on the minimum point distance of IoU—enhances bounding box regression training, leading to improved convergence speed and regression accuracy. Experimental results show that YOLOv8_DSM achieves superior detection accuracy, with a mAP of 63.4%, and maintains high real-time performance (144.6 FPS), marking a 5.14% improvement over the original model. This meets the industrial requirements for PCB defect detection in terms of both accuracy and real-time efficiency.

Although YOLOv8_DSM performs well in detecting various defect types, certain limitations persist under specific conditions. Specifically, factors such as lighting variations and overlapping defects can affect detection accuracy, leading to suboptimal results for small or complex defects. To enhance robustness and generalization, future research could explore integrating multimodal data fusion, applying lighting compensation techniques, and training on more diverse datasets to better address these challenging defect detection scenarios.

## Figures and Tables

**Figure 1 sensors-24-07373-f001:**
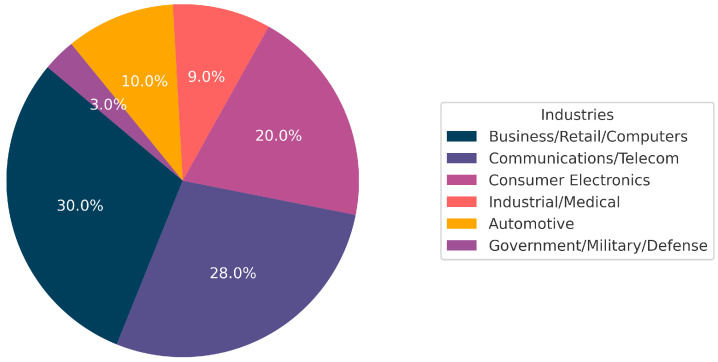
Proportions of PCB Applications in Various Fields.

**Figure 2 sensors-24-07373-f002:**
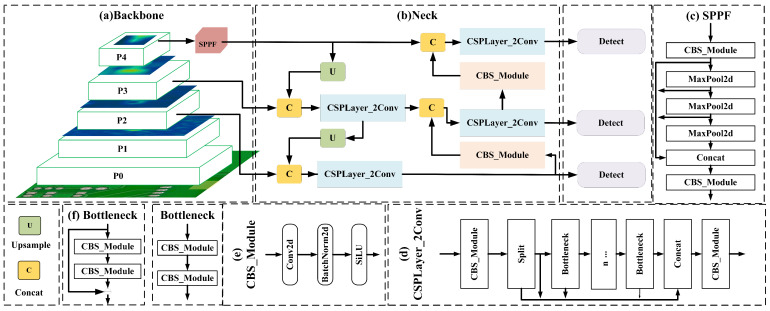
A detailed network architecture diagram of YOLOv8n, where (**a**) represents the backbone of the network, (**b**) outlines the structure of the network’s Neck, (**c**) illustrates the detailed structure of the SPPF module, (**d**) showcases the detailed structure of the CSPLayer_2Conv, (**e**) displays the detailed diagram of the CBS convolutional model, and (**f**) presents two detailed structures of the Bottleneck.

**Figure 3 sensors-24-07373-f003:**
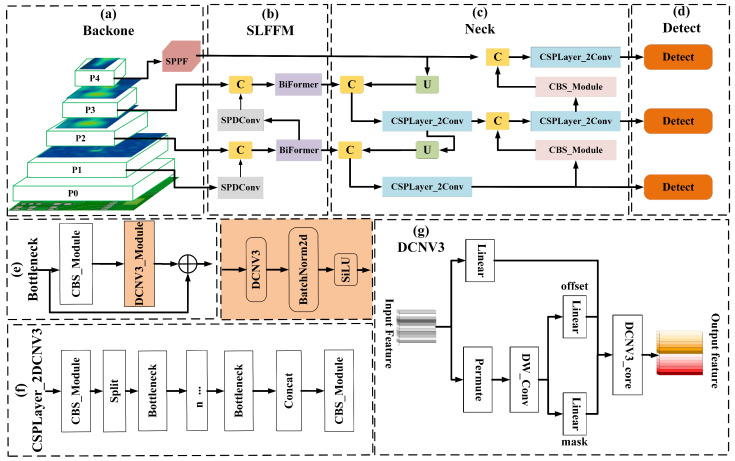
A detailed YOLOv8_DSM network architecture diagram, where (**a**) represents the network’s backbone, (**b**) illustrates the detailed structure of the SLFFM, (**c**) is the network’s Neck, (**d**) denotes the network’s detection head, (**e**) showcases the detailed structure of the network’s Bottleneck, (**f**) presents the detailed structure of the CSPLayer_2Dcnv3, and (**g**) elucidates the detailed structure of the DCNV3 convolution.

**Figure 4 sensors-24-07373-f004:**
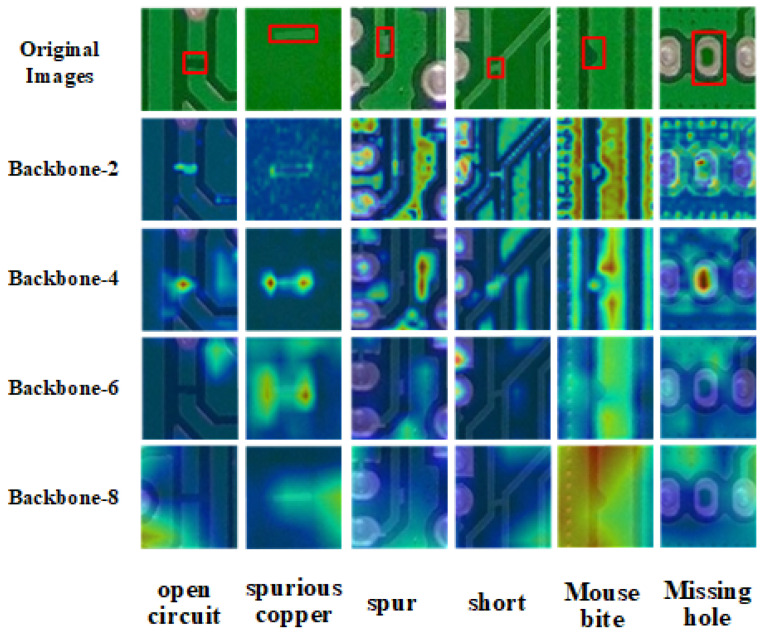
Visualization of Heatmaps for Different Layer Feature Maps in the backbone, with red boxes indicating the location of defects.

**Figure 5 sensors-24-07373-f005:**
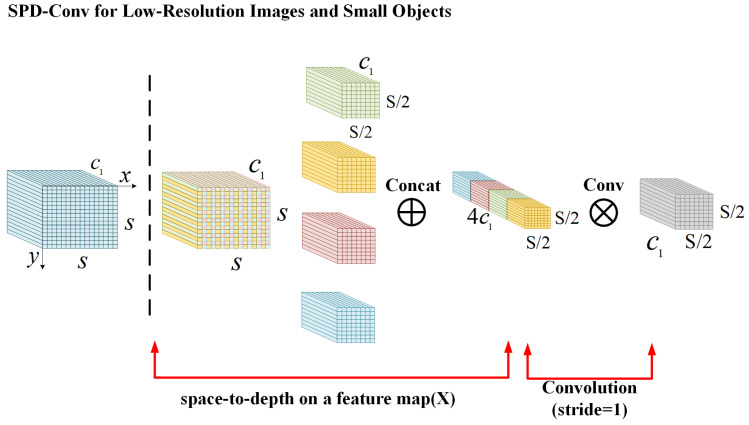
Illustration of SPD-Conv when scale equals to two.

**Figure 6 sensors-24-07373-f006:**
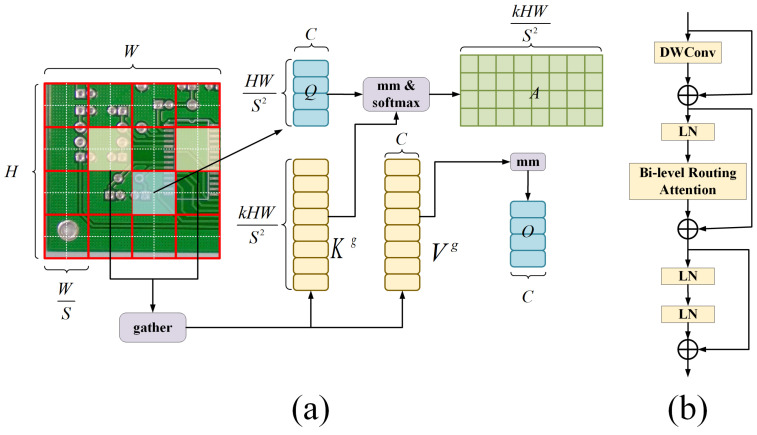
(**a**) Structure of the Bi-Level Routing Attention; (**b**) Structure of the BRA.

**Figure 7 sensors-24-07373-f007:**
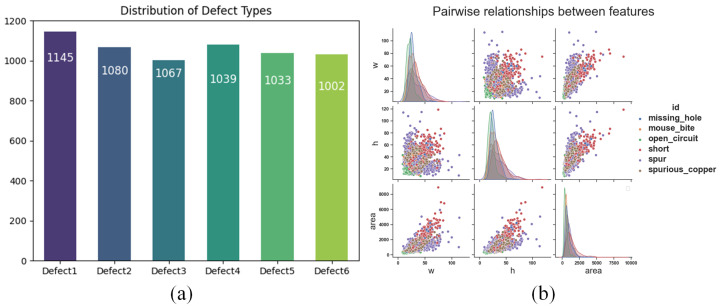
Depicts the statistical information of the dataset. Where (**a**) represents the number of various defects in the dataset, and (**b**) represents the distribution of width, height, and area of various defects in the dataset.

**Figure 8 sensors-24-07373-f008:**
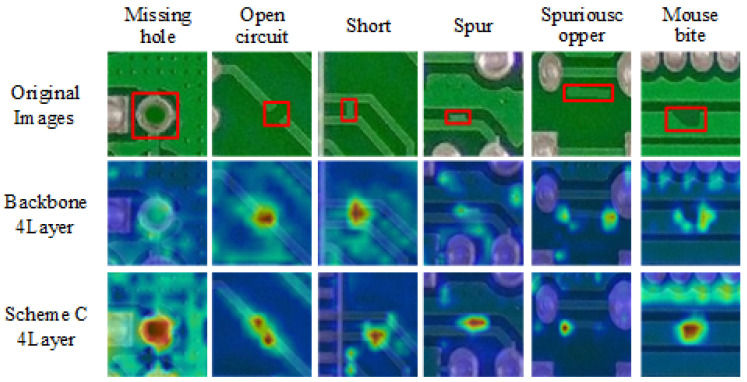
Heatmap visualization of the fourth layer feature map in the original backbone and Scheme C, with red boxes indicating the location of defects.

**Figure 9 sensors-24-07373-f009:**
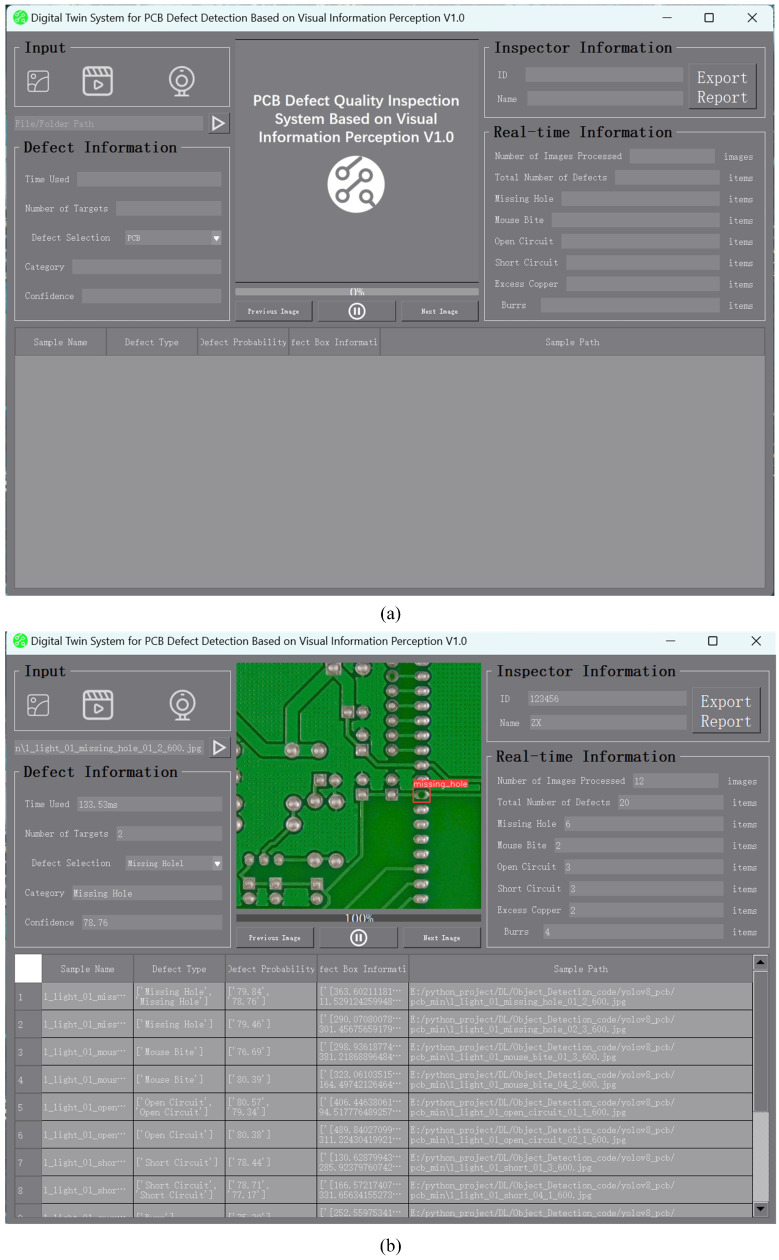
Main interface of the PCB defect detection quality inspection system: (**a**) is the initial state, and (**b**) is the running state.

**Table 1 sensors-24-07373-t001:** Performance comparison table of YOLOv5n to YOLOv8n target detection algorithms at COCO2017.

Model	Size (Pixels)	Parameters (M)	FLOPs (G)	mAP (0.5:0.95)
YOLOv5n	640	1.9	4.2	28.0
YOLOv6n	640	4.7	11.4	37.5
YOLOv7-tiny	640	6.2	13.1	36.8
YOLOv8n	640	3.2	8.7	37.3

**Table 2 sensors-24-07373-t002:** Experimental software and hardware environment parameter table.

Parameter	Configuration
GPU	NVIDIA GeForce RTX 4090
Operation System	Ubuntu20
Deep learning architecture	Pytorch1.11.0
Pre-trained weight	YES
Batch size	32
Input size	640 × 640
Optimizer	SGD
Learning rate Weight	0.02
Decay Momentum	0.0005
momentum	0.937
EarlyStopping patience	Yes
Data enhancement strategy	Mosaic

**Table 3 sensors-24-07373-t003:** Performance Comparison of Model Test Results.

Model	Parameters (M)	mAP (0.5:0.95)	Inference Time (ms)
Yolov5n [40]	**1.8**	55.2%	2.3 ^1^
Yolov6n [41]	4.6	60.3%	1.5 ^1^
Yolov7-tiny [42]	6	59.5%	2.2 ^1^
Yolov8n [30]	3.0	60.3%	2.1 ^1^
YOLO-MBBi [25]	8.8	52.5%	79.2 ^2^
GCC-YOLO [43]	8.2	48.5%	70.2 ^2^
Light-PDD [44]	3.84	47.1%	109.2 ^2^
YOLO-HMC [29]	5.94	54.9%	44.6 ^2^
Ours	3.8	**63.4%**	3.5 ^1^

^1^ Experimental data obtained using a server equipped with an NVIDIA GeForce RTX 4090 graphics card. ^2^ Data from Reference [29], obtained using a server equipped with an NVIDIA GeForce GTX 1650 Ti. Bolded numbers represent optimal performance indicators.

**Table 4 sensors-24-07373-t004:** Model experimental results using different backbones, class1 is missing_hole, class2 is mouse_bite, class3 is open_circuit, class4 is short, class5 is spur, class6 is spurious_copper.

Method	Class1	Class2	Class3	Class4	Class5	Class6	All	Parameters (M)	FLOPs (G)
Backbone	61.9%	61.1%	55.6%	64.4%	**59.4%**	59.2%	60.3%	3.0	8.1
Scheme A	59.5%	58.7%	52.3%	59.5%	54.6%	56.5%	56.8%	2.8	**7.6**
Scheme B	59.7%	59.3%	51.6%	59.9%	54.8%	57%	57%	2.9	7.7
Scheme C	**63.9%**	**63.2%**	**55.6%**	**64.4%**	56.8%	**59.8%**	**60.6%**	**2.8**	7.7

Bolded numbers represent optimal performance indicators.

**Table 5 sensors-24-07373-t005:** Statistical table of ablation experiment results for shallow low-semantic multi-scale fusion modules, class1 is missing_hole, class2 is mouse_bite, class3 is open_circuit, class4 is short, class5 is spur, class6 is spurious_copper.

SPDConv	BRA	Class1	Class2	Class3	Class4	Class5	Class6	All
	✔	66.3%	**64.3%**	**60.7%**	64.1%	59.1%	61%	62.6%
✔		64.6%	64.1%	57%	64.8%	59.6%	60.3%	61.8%
✔	✔	**66.7%**	64.1%	59.3%	**65%**	**60.2%**	**61.5%**	**62.8%**

Bolded numbers represent optimal performance indicators.

**Table 6 sensors-24-07373-t006:** Ablation experiment results for each module of the YOLOv8_DSM model.

Yolov8	DCNv3	SLFFM	MPDIoU	mAP (0.5:0.95)	Parameters (M)	FLOPs (G)
✔				60.3%	3.0	8.1
✔	✔			60.7%	**2.8**	**7.6**
✔		✔		62.8%	4	11.5
✔	✔	✔		63.1%	3.8	11.1
✔	✔	✔	✔	**63.4%**	3.8	11.1

Bolded numbers represent optimal performance indicators.

## Data Availability

The original contributions presented in the study are included in the article, further inquiries can be directed to the corresponding author.

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
