# Peer review of "An Optimization Method for PCB Surface Defect Detection Model Based on Measurement of Defect Characteristics and Backbone Network Feature Information"

_sensors, 2024, doi:10.3390/s24227373_

Round 1
Reviewer 1 Report
Comments and Suggestions for Authors
The paper titled " An Optimization Method for PCB Surface Defect Detection Model Based on Measurement of Defect Characteristics and Backbone Network Feature Information" presents proposes the YOLOv8_DSM algorithm for PCB surface defect detection, aiming to address detection challenges, including diverse and complex features, low feature resolution, and similarities to the background. Through extensive testing on various benchmarks for PCB surface defect detection, the effectiveness of this approach is demonstrated. Finally, the proposed model is deployed in a PCB quality inspection system.
Overall, I had a positive experience reading the paper. It is well-structured and clearly written. However, I have identified a few areas where clarity could be enhanced:
The description of the selected models for comparison (as shown in Table 3) needs clarification. Please add a brief explanation highlighting the reason behind choosing these specific models for comparison.
I noticed the comparison of the number of parameters and detection accuracy among the different models (in Table 3). To better illustrate real-time performance, it would be beneficial to include the detection time for each model.
The explanation of Table 4 requires improvement. The authors should clarify why Scheme C shows a decrease in average accuracy for excess copper. Providing a deeper analysis of the underlying reasons will enhance readers’ understanding.
Author Response
Dear Reviewer,
Thank you for taking the time and effort to review our manuscript and for providing invaluable suggestions. This will greatly contribute to the publication of our manuscript. We have carefully read all your comments and have made conscientious revisions to our manuscript based on your recommendations to enhance its readability and professionalism. In the remainder of this letter, we will address your comments individually and provide corresponding responses. For clarity in our discussion, we will first restate your comments in italics and then respond to each one.
Comments 1:
The description of the selected models for comparison (as shown in Table 3) needs clarification. Please add a brief explanation highlighting the reason behind choosing these specific models for comparison.
Response 1:
Thank you for your insightful suggestion regarding the need for detailed explanations on the selection of comparative models in our manuscript. In response, we have enriched the first paragraph of Section 3.3.1 with relevant content: “Yolov5n, Yolov6n, Yolov7-tiny, and Yolov8n are classic, high-performing single-stage algorithms widely used as baseline models, while YOLO-MBBi, GCC-YOLO, Light-PDD, and YOLO-HMC are prominent current algorithms recognized for their exceptional performance in PCB defect detection tasks.” This addition provides a clear rationale for our model choices, enhancing readers’ understanding of our selections.
Comments 2:
I noticed the comparison of the number of parameters and detection accuracy among the different models (in Table 3). To better illustrate real-time performance, it would be beneficial to include the detection time for each model.
Response 2:
Thank you for your valuable suggestion regarding the inclusion of real-time performance metrics in Table 3. Based on your feedback, we have added this metric to the table. However, we would like to highlight that there are significant differences in the real-time performance metrics due to the experimental data being generated on two servers with different configurations. We have noted this fact in the footnote of the table. Specifically, the models YOLOv5n-v8n and our designed model were trained on a server equipped with an NVIDIA GeForce RTX 4090, while YOLO-MBBi, GCC-YOLO, Light-PDD, and YOLO-HMC were obtained from Reference [1], utilizing a dataset identical to ours, but implemented on a server with an NVIDIA GeForce GTX 1650 Ti. Therefore, we initially included only the model size and accuracy metrics in the table, as these indicators are not dependent on the server configuration.
Reference:
[1] Yuan, M.; Zhou, Y.; Ren, X.; Zhi, H.; Zhang, J.; Chen, H. YOLO-HMC: An Improved Method for PCB Surface Defect Detection. IEEE Transactions on Instrumentation and Measurement 2024, 73, 1–11. https://doi.org/10.1109/TIM.2024.3351241.
Comments 3:
The explanation of Table 4 requires improvement. The authors should clarify why Scheme C shows a decrease in average accuracy for excess copper. Providing a deeper analysis of the underlying reasons will enhance readers’ understanding.
Response 3:
Thank you for your constructive feedback on the need to enhance explanations in Table 4 of our manuscript. In response, we have added a detailed analysis to the second paragraph of Section 3.3.2: “Scheme C exhibited a minor decrease in average precision for spur defects, likely attributed to a reduced capacity for fine-detail feature extraction due to a lower parameter count and computational complexity.” This explanation clarifies the underlying reason for the observed accuracy drop in Scenario C, thereby providing readers with a deeper understanding of the model’s performance nuances.
Sincerely,
The Authors
Reviewer 2 Report
Comments and Suggestions for Authors
The manuscript presents an optimized approach for detecting surface defects in Printed Circuit Boards (PCBs) by proposing a new algorithm, YOLOv8_DSM. PCB surface defects are challenging to detect due to their low resolution, background similarity, and varied shapes. The YOLOv8_DSM model introduces three main improvements: (1) a modified backbone network with deformable convolution layers (CSPLayer_2DCNv3) for adaptive feature extraction, (2) a Shallow-layer Low-semantic Feature Fusion Module (SLFFM) to enhance low-resolution feature representation and mitigate background similarity issues, and (3) an MPDIoU bounding box loss function for improved training efficiency. Experimental results demonstrate that YOLOv8_DSM achieves a mean Average Precision (mAP) of 63.4% (a 5.14% improvement over the original YOLOv8 model) and a frame rate of 144.6 FPS, making it suitable for real-world PCB inspection systems.
The paper addresses a significant industrial problem—defect detection in PCBs, which is critical for quality assurance in electronics manufacturing. The proposed YOLOv8_DSM algorithm shows potential for practical application, as demonstrated by its deployment on a PC-based quality inspection system. The inclusion of CSPLayer_2DCNv3 with deformable convolutions to capture complex defect shapes is a novel approach, showing an understanding of the diverse and irregular nature of PCB defects. Achieving an mAP of 63.4% and a frame rate of 144.6 FPS highlights the model’s suitability for real-time, high-throughput defect detection, making it valuable for industrial applications.
Please find some comments and suggestions below:
While the paper introduces the SLFFM and CSPLayer_2DCNv3 modules, it would benefit from a more detailed explanation of how these modules work. Specifically, a breakdown of the SLFFM module’s design, its role in feature fusion, and how it mitigates defect-background similarity would enhance understanding.
It is suggested to introduce more recent work on the application of lightweight CNN networks for defect detection problems, such as: https://doi.org/10.1007/s00170-022-10335-8
The paper compares YOLOv8_DSM with the original YOLOv8, but it would benefit from a broader comparison with other state-of-the-art PCB defect detection models, such as Faster R-CNN, EfficientDet, or RetinaNet.
An ablation study showing the individual contributions of each component (e.g., CSPLayer_2DCNv3, SLFFM, BRA mechanism, MPDIoU) would strengthen the manuscript. Presenting the mAP and FPS of the model with and without these components would clarify the impact of each design choice on the overall performance.
An error analysis discussing the model’s limitations in detecting certain defect types or under specific conditions (e.g., lighting variations, overlapping defects) would add depth to the evaluation. Are there any common failure cases or types of defects that remain challenging for YOLOv8_DSM?
Overall, this manuscript presents a valuable contribution to the field of PCB defect detection, offering a targeted approach to address the unique challenges of low-resolution, background-similar defects. The proposed YOLOv8_DSM algorithm demonstrates significant improvements in accuracy and speed, making it well-suited for industrial applications.
Author Response
Dear Reviewer,
Thank you for taking the time and effort to review our manuscript and for providing invaluable suggestions. This will greatly contribute to the publication of our manuscript. We have carefully read all your comments and have made conscientious revisions to our manuscript based on your recommendations to enhance its readability and professionalism. In the remainder of this letter, we will address your comments individually and provide corresponding responses. For clarity in our discussion, we will first restate your comments in italics and then respond to each one.
Comments 1:
While the paper introduces the SLFFM and CSPLayer_2DCNv3 modules, it would benefit from a more detailed explanation of how these modules work. Specifically, a breakdown of the SLFFM module’s design, its role in feature fusion, and how it mitigates defect-background similarity would enhance understanding.
Response 1:
Thank you for your interest in the design of our SLFFM module, its role in feature fusion, and its effectiveness in reducing similarity between defects and their backgrounds. To enhance readers' comprehension of this innovation, we have expanded the explanation and provided a more detailed description in Section 2.3.2, with key points highlighted for emphasis.
Comments 2:
It is suggested to introduce more recent work on the application of lightweight CNN networks for defect detection problems, such as: https://doi.org/10.1007/s00170-022-10335-8
Response 2:
Thank you for your attention to the research background of our manuscript. We carefully reviewed the reference you provided. WearNet, a lightweight convolutional neural network, achieved an impressive 94.16% classification accuracy in scratch detection, representing a highly significant research contribution. We have cited this recommended reference in Section 1, "Introduction," to enhance the relevance and depth of our study within the field.
Comments 3:
The paper compares YOLOv8_DSM with the original YOLOv8, but it would benefit from a broader comparison with other state-of-the-art PCB defect detection models, such as Faster R-CNN, EfficientDet, or RetinaNet.
Response 3:
Thank you for your insightful comments regarding the selection of comparative models in our manuscript. In response, we have provided additional explanations in Section 3.3.1 to clarify our model selection criteria. While Faster R-CNN, EfficientDet, and RetinaNet are indeed widely recognized models, our study specifically focuses on comparing single-stage object detection algorithms, particularly the YOLO series, due to their proven efficiency and suitability for real-time PCB defect detection tasks. This focus allows for a more targeted evaluation of single-stage models in this application context, aligning with the objectives of our research.
Comments 4:
An ablation study showing the individual contributions of each component (e.g., CSPLayer_2DCNv3, SLFFM, BRA mechanism, MPDIoU) would strengthen the manuscript. Presenting the mAP and FPS of the model with and without these components would clarify the impact of each design choice on the overall performance.
Response 4:
Thank you for your attention to the ablation experiments in our manuscript. Our main innovations in this study include the design of CSPLayer_2DCNv3 and SLFFM, as well as the use of an improved MPDIoU loss function. Accordingly, we have conducted detailed ablation studies on these three innovations in Table 6 to clarify the unique contributions of our work. Additionally, in Table 5, we examine the specific effects of the SPDconv and BRA mechanisms on SLFFM performance. Therefore, the points you mentioned are thoroughly addressed and described in detail in our manuscript.
Comments 5:
An error analysis discussing the model’s limitations in detecting certain defect types or under specific conditions (e.g., lighting variations, overlapping defects) would add depth to the evaluation. Are there any common failure cases or types of defects that remain challenging for YOLOv8_DSM?
Response 5:
Thank you for your constructive feedback on our evaluation of certain defect types and specific conditions in our manuscript. We have incorporated additional insights into the outlook and analysis in the second paragraph of Section 5, which will help provide relevant perspectives and directions for future research.
Sincerely,
The Authors